# Potential Celiac Disease in Children: Health Status on A Long-Term Gluten-Containing Diet

**DOI:** 10.3390/nu16111708

**Published:** 2024-05-30

**Authors:** Roberta Mandile, Federica Lerro, Martina Carpinelli, Lorenzo D’Antonio, Luigi Greco, Riccardo Troncone, Renata Auricchio

**Affiliations:** 1Department of Translational Medical Sciences, University of Naples Federico II, 80131 Naples, Italy; 2European Laboratory for the Investigation of Food Induced Disease (ELFID), University Federico II, 80131 Naples, Italy

**Keywords:** potential celiac disease, gluten-free diet, nutritional status, autoimmunity complications, anti-tissue transglutaminase antibody

## Abstract

Potential celiac disease (PCD) is a clinical condition characterised by the presence of a positive CD-specific serology and a normal intestinal architecture. Asymptomatic PCD patients are generally advised to continue on a gluten-containing diet (GCD), but long-term risks of this approach have never been explored. In the present study, we aimed to investigate nutritional and autoimmune complications possibly developing overtime in a cohort of asymptomatic PCD children on a GCD. We compared children’s parameters of growth, nutritional status, and autoimmunity between the time of diagnosis and on the occasion of their last medical check, after a long-term gluten-containing diet. Altogether, we collected data from 171 PCD children with a mean follow-up time of 3 years (range 0.35–15.3 years). During follow-up, although patients did not reduce their amount of daily gluten intake, their anti-tissue transglutaminase (anti-TG2) antibodies spontaneously and significantly decreased. Most parameters analysed had not changed during follow-up (height centile, ferritin, albumin, cholesterol, calcium, alkaline phosphatase, parathormone, and vitamin D) or even improved significantly (weight and BMI centile, haemoglobin, blood iron, HDL, glycaemia, and HbA1C, *p* < 0.05), always remaining within the limit of normality. Equally, autoantibodies for other concomitant autoimmune disorders did not increase overtime. Similar results were obtained excluding from analysis patients who had stopped producing anti-TG2 and those with a follow-up time < 3 years. Our pilot study has provided reassuring results regarding the maintenance of a gluten-containing diet in asymptomatic PCD children, even when long-term follow-up was considered.

## 1. Introduction

Celiac disease (CD) has been defined by the ESPGHAN as a systemic immune-mediated disorder elicited by gluten and related prolamines in genetically susceptible individuals [1]. It is characterised by the presence of a variable combination of clinical manifestations, CD-specific autoantibodies, HLA-compatible haplotypes, and enteropathy. This histological alteration can range from a completely altered mucosa to one with normal architecture, with a preserved villous-to-crypt ratio, with or without an increased number of intraepithelial lymphocytes (IELs). The latter condition is defined as potential celiac disease (PCD) [2]; it is characterised by a positive CD-associated serology without intestinal damage [3]. The implementation of screening strategies for at-risk groups, as well as increased attention to subtle signs of the disease, has led in recent years to a more liberal use of serological diagnostic tools and increased detection of these subjects. Globally, potential CD (PCD) is estimated to be around 10% of the total CD diagnosis, but the range is quite wide, from 6% to 30% [4]. Its clinical management is variable due to the lack of sufficient knowledge of its natural history. Some recent studies, mostly undertaken using adult patients, suggest that gluten-dependent symptoms may be present even without villous atrophy (VA). They also show that most patients evolve to overt CD when continuing on a gluten-containing diet (GCD) [5,6,7,8,9]. On the contrary, there is increasing evidence that in children PCD is often an asymptomatic condition, and evolution to VA is not the rule [10,11,12,13]. Some patients even stop producing autoantibodies despite being on a GCD, suggesting, in these cases, a reversibility of the autoimmune process [12]. Specifically, a recent study from our group demonstrated that the cumulative incidence of progression to villous atrophy was 43% over a 12-year period, suggesting that more than half of patients on a gluten-containing diet remained asymptomatic and histologically healthy over long-term follow-up [12]. Based on these data, the scientific community, supported by the latest European Society of Paediatric Gastroenterology Haepatology and Nutrition (ESPGHAN) guidelines on CD follow-up in children, suggests a gluten-containing diet to asymptomatic PCD patients, reserving recommending a gluten-free diet (GFD) for symptomatic patients [3]. Unfortunately, the issue of whether chronic gluten consumption may induce clinical or even subclinical complications on children’s nutritional status over time, as well as the appearance of other concomitant autoimmune diseases, is poorly studied in the literature.

Taking advantage of the opportunity to study different aspects of PCD natural history in our prospective cohort of PCD children [12], we designed a retrospective study with the primary aim of understanding whether, in a group of PCD patients that remained asymptomatic during follow-up, long-term chronic gluten exposure can somehow alter, even in a subclinical way, their growth or nutritional status. The secondary aim was to assess an eventual increase in other autoimmune disorders during the follow-up of PCD patients on a gluten-containing diet.

## 2. Materials and Methods

We performed an observational study selecting from the whole cohort of PCD children those persistently asymptomatic on a gluten-containing diet. 

Patients were classified as potential celiac if they tested positive twice for anti-transglutaminase (anti-TG2) antibodies, which was confirmed by anti-endomysial antibodies (EMA), and had a normal or just-infiltrated intestinal mucosa (Marsh stages 0 or 1). All had HLA-DQ2 and/or -DQ8 haplotypes. Total serum IgA antibodies were within the normal range.

Our whole PCD cohort included 515 patients enrolled since 2000: 72 immediately started a GFD because of the presence of symptoms, and 443 were prospectively followed-up, with a mean follow-up time of 45 months (maximum 17 years). During follow-up, clinical visits were scheduled and blood samples were obtained every 6–12 months, while a duodenal biopsy was performed every 2 years, or earlier for patients with symptoms. The amount of gluten intake was evaluated by a dedicated nutritionist based on 3-day food records and expressed as grams of gluten taken per day. During the follow-up, 124/443 (28%) started a gluten-free diet (GFD) because of the development of VA (15%) or relevant symptoms without a new duodenal biopsy.

Among the 319 patients with PCD that did not develop symptoms or VA, we were able to retrospectively collect detailed clinical nutritional and autoimmune data for 171 (53.6%) patients (see flow chart in Figure 1).

For these patients we had the following data: growth parameters (height, weight, body mass index, and relative percentiles matched for age and sex), serum levels of anti-TG2, albumin, iron metabolism (haemoglobin, ferritin, and serum iron), bone metabolism (calcium, phosphorus, alkaline phosphatase, vitamin D, and parathormone), glucose metabolism (fasting glycaemia and HbA1C), lipid profile (LDL, HDL, and triglycerides), autoantibodies for thyroid (anti-TG and anti-TPO), type 1 diabetes (anti-IAA, anti-IA2, anti-Zn-T8, and anti-GAD), and ANA.

We compared patients’ parameters at the time of diagnosis (when the first duodenal biopsy was performed) and after a long-term gluten-containing diet on the occasion of their last medical check. We excluded from the analysis patients that supplemented with micronutrients such as iron or vitamin D. As the cohort also included patients that had permanently stopped producing antibodies over time (repeated negative determinations in serum for at least 2 years, N = 37), we performed a second analysis excluding them, to ascertain if a possible nutrition impairment may have been limited to patients who were persistently producing anti-TG2. Similarly, to avoid the possible bias related to a relatively short follow-up, we performed another sub-analysis only considering patients with a follow-up longer than 3 years (N = 75). Parameters were compared using a paired t-test for normally distributed variables and a contingency table with a *Chi square* test for qualitative variables. Statistical analysis was performed using SPSS software ver. 27.1. This study was approved by the ethical committee of University Federico II.

## 3. Results

### 3.1. Whole Cohort

Of 171 PCD patients, 60% were female, the mean age at diagnosis was 6 years (range 1.6–17 years), and 36% reported in their history familiarity for CD. All were EMA positive, and Marsh 0 and Marsh 1 lesions were present in 50/50%, respectively. Patients on a gluten-containing diet were followed up for a mean time of 3 years (range 0.35–15.3 years) (Table 1). No one received iron supplementation, and 6/171 patients (3.5%) received vitamin D supplementation after diagnosis; of course these six patients were excluded from the analysis aimed to evaluate vitamin D changes over time, as their exogenous administration might have biased the analysis. No significant difference was noted between the amount of gluten per day at diagnosis and at the last follow-up (mean 23 gr gluten/day, SD 11 at diagnosis vs. 21 gr/day, SD 10.8 at last follow-up, *p* = 0.3). Their anti-TG2 antibodies significantly decreased from 3.7 times the upper limit of normality at diagnosis to 1.8 at the last follow-up. Significant increases in BMI (mean at time 0 vs. time at last follow-up: 16.4 kg/m^2^, SD 2.3 vs. 17.89 kg/m^2^, SD 3.5, respectively, *p* < 0.0001), BMI percentile (48.8 centile, SD 32 vs. 53.0 centile, SD 33, respectively, *p* = 0.04) and weight percentile (46.8 centile, SD 31.3 vs. 51.3 centile, SD 30.1, respectively, *p* = 0.01) were observed. We also noted significant increases in haemoglobin (12.8 gr/dL, SD 0.9 vs. 13.3 gr/dL, SD 1, respectively, *p* < 0.0001), serum iron (81.0 µg/dL, SD 31.9 vs. 88.9 µg/dL, SD 38.3, respectively, *p* = 0.03), HDL (54.2 mg/dL, SD 12.1 vs. 56.4 mg/dL, SD 12.4, respectively, *p* = 0.02), basal glucose (67.1 mg/dL, SD 9.8 vs. 69.7 mg/dL, SD 8.6, respectively, *p* = 0.003), and HbA1C (5.21%, SD 0.25 vs. 5.32%, SD 0.27, respectively, *p* = 0.01). All parameters remained within the limit of normality at any time. Equally, we noted significant decreases in levels of calcium (9.84 mg/dL, SD 0.40 vs. 9.74 mg/dL, SD 0.42, respectively, *p* = 0.03) and phosphorus (4.8 mg/dL, SD 0.61 mg/dL vs. 4.5 mg/dL, SD 0.64, respectively, *p* = 0.0002). Also, in this case all parameters remained within the limit of normality, and vitamin D, parathormone, and alkaline phosphatase levels did not significantly change. All other parameters analysed did not significantly change. We scrutinized individual patients on a gluten-containing diet for variables indicative of nutritional status. Of these, 3/171 (1.75%) had abnormal haemoglobin values for their age and sex at the end of their follow-up: one had a known thalassemia tract, in one haemoglobin levels improved compared to the time of diagnosis, and in one remained comparable (with an improved ferritin value). In addition, 18/171 (10%) patients had values of BMI < 10 percent for their age and sex, in 10 patients it remained stable or even improved at the end of follow-up, and in 8 patients (4.6% of total) BMI decreased. Hypoalbuminemia did not develop (Figure 2).

There was not a statistically significant change in the number of patients that presented at least one positive autoantibody for thyroiditis (anti-TG or anti-TPO, *p* = 0.06) or type 1 diabetes (anti-IAA, anti-IA2, anti-Zn-T8, or anti-GAD, *p* = 0.15). ANA were detected in 23 patients at the end of the period of follow-up, but in no cases at diagnosis.

### 3.2. Subgroup Analysis: Excluding Patients That Permanently Stopped Producing Antibodies during Follow-Up and Excluding Patients with a Follow-Up < 3 Years

We performed second and third analyses with the same procedures excluding, respectively, patients that permanently stopped producing antibodies during follow-up and patients with a follow-up < 3 years. Results overlapped with those obtained from the whole cohort. Detailed results are reported in Table 2.

## 4. Discussion

Despite numerous efforts recently spent on the aim of finding alternative strategies, a GFD remains the cornerstone of CD therapy. It is safe and effective in both addressing clinical symptoms and restoring a normal intestinal architecture, the latter being the final goal of CD therapy [14]. Additionally, it must be considered that only one randomised clinical trial has clearly demonstrated benefits of a GFD when adopted in asymptomatic patients with atrophic mucosa [15]. The role of a gluten-free diet is even less certain in a condition like PCD, where intestinal mucosa are already architecturally healthy in the presence of a positive CD-associated serology. In addition, prescribing an inappropriate GFD is not a safe option, as it can induce nutritional deficiencies, body weight gain, and important social and psychological impairments [16]. Current ESPGHAN guidelines suggest starting a trial GFD in symptomatic patients, despite its having been shown that gastrointestinal symptoms in PCD are not always gluten dependent and may not improve with such a diet [17]. Asymptomatic PCD patients are even more challenging. Indeed, they include a heterogeneous group of patients who may eventually develop VA or clinical symptoms at a later date, but more frequently continue producing autoantibodies without any intestinal damage or even permanently stop producing them. Many studies have tried to identify risk factors able to predict, at the time of diagnosis, the future evolution of PCD patients. We previously found that an older age at diagnosis and an increased number of IELs (especially those expressing on their surface γδ receptors) together with an individual genetic predisposition (that includes both HLA and non-HLA genes) are sufficient to correctly predict, at entry, 80% of children who will not develop a flat mucosa during follow-up, and approximately 69% of those who will develop flat mucosa [12]. These data, however, cannot represent more than a guide for the clinician, and ESPGHAN recommendations currently do not suggest a GFD in asymptomatic patients whatever their risk class, even if they encourage referring PCD patients for follow-up in an experienced tertiary care centre. These same guidelines highlight in the conclusion section the necessity to study long-term risks eventually developing in asymptomatic PCD patients maintained on a gluten-containing diet [14]. In this context, we took advantage of our consistent cohort of PCD patients prospectively monitored on a gluten-containing diet to assess if, in the subgroup of patients clinically asymptomatic and without intestinal damage, chronic gluten exposure impacted their nutritional status or the development of other autoimmune disorders. We designed a retrospective observational study to compare, in each patient, different parameters at diagnosis and after a long period on a GCD.

It is well known that in atrophic CD patients the maintenance of a gluten-containing diet easily induces progressive weight loss and biochemical signs of malnutrition such as anaemia, and in more serious cases hypoalbuminemia linked to the concomitant enteropathy.

Vice versa, we did not find a worsening in the growth, iron metabolism, glycaemic metabolism, or nutritional status in our cohort of PCD patients. We found that some parameters even improved (like BMI, haemoglobin, serum iron, basal glycaemia, and HbA1C), always remaining within the limit of normality. We surprisingly found a significant reduction in levels of serum calcium and phosphorus, but the stability of alkaline phosphatase levels, parathormone levels, and vitamin D levels during follow-up did not support the hypothesis of a concomitant bone remodeling. This can be explained by the fact that we collected data from a large cohort of patients, and data show a Gaussian distribution. In this data set, even small changes reached statistically significant relevance, but this does not mean that they have a concomitant clinical relevance. Also, performing a detailed analysis of single patients for the most important variables, we did not identify patients that became anaemic or hypoalbuminemic during follow-up. As our cohort also included patients that permanently stopped producing anti-TG2 antibodies, we wondered whether nutritional complications occurred only in the subgroup of patients that continued to produce autoantibodies during follow-up. This does not seem to have been the case, as in this group, we had just-overlapping results compared to the whole cohort. We also evaluated how circulating autoantibodies, markers of other autoimmune disorders known to be associated to CD, increased during follow-up of patients on a gluten-containing diet. Minimal data are available on this topic in the literature: only one study of adult PCD, in fact, seems to suggest that PCD patients kept on a GCD have an increased risk to develop other concomitant autoimmune disorders [18]. This was not the case in this study, as during follow-up we did not find an increase in autoantibodies for thyroiditis or type one diabetes, which are the most important autoimmune pathologies known to be associated to CD. Moreover, we are aware that these antibodies, without clinical symptoms, only represent a marker of autoimmune predisposition in patients that are genetically predisposed to autoimmunity, and the causative role of gluten cannot be elucidated in a study design like ours.

This study presented some limitations. The best way to verify the effect of a GFD on PCD patients is a double-blind randomised clinical trial with a sufficiently long follow-up time. In this context, quality of life should also be investigated through standardised questionnaires. In our case, we performed a retrospective observational study, not randomised, and without a real control group of asymptomatic PCD patients maintaining a GFD. We also tried to overcome the bias of follow-up duration, as nutritional and autoimmune complications need time to become detectable. We performed an analysis on only the subgroup of patients with a follow-up longer than 3 years, and in this case, we found overlapping results compared to the whole cohort. Finally, one can argue that we excluded from the analysis just those patients whose condition worsened, developing symptoms or VA. We should consider that in these cases, based on current recommendations, patients start a GFD anyway, whereas we wanted to answer the question of what happens to those patients that we normally follow-up who maintain a gluten-containing diet, to be sure not to cause them damage, even if subclinical.

## 5. Conclusions

In conclusion, our pilot study provided reassuring results regarding the maintenance of a gluten-containing diet in the subgroup of PCD patients that were clinically asymptomatic during follow-up, as it seems that their nutritional and autoimmune statuses were not impaired, even when a long follow-up was considered. Further studies are requested to validate these data on a larger cohort of individuals, ideally through designing and performing a double-blind randomised clinical trial. This may also help correctly investigate the role of gluten in the induction of other concomitant autoimmune disorders in individuals with a predisposing genetic background.

## Figures and Tables

**Figure 1 nutrients-16-01708-f001:**
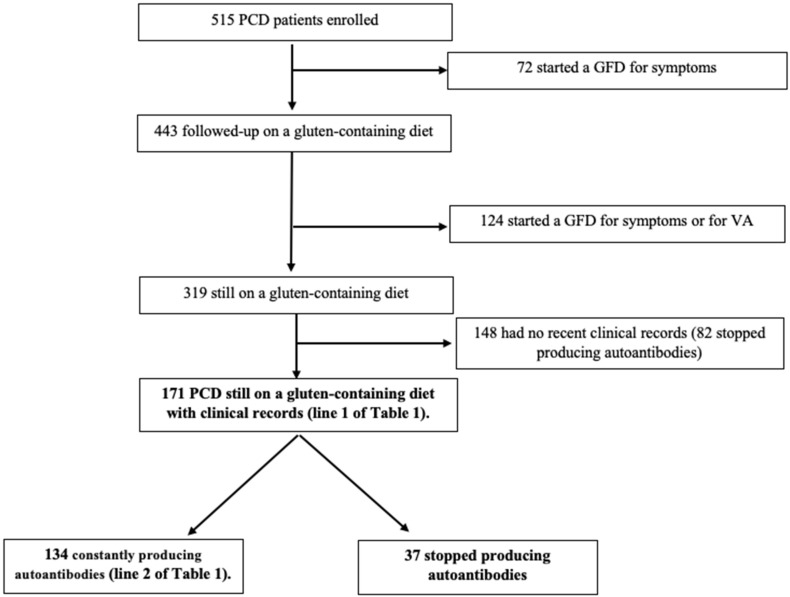
Flow chart of study cohort. GFD: gluten-free diet. VA: villous atrophy.

**Figure 2 nutrients-16-01708-f002:**
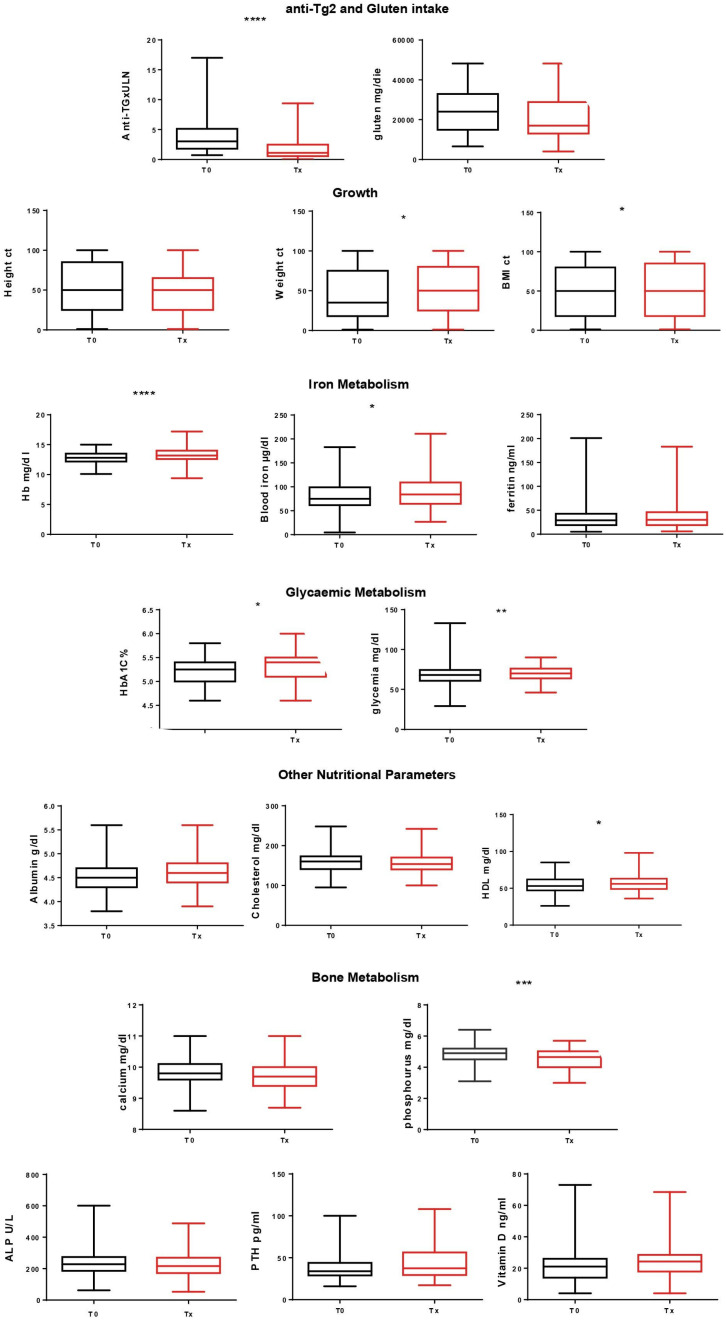
Parameters between time of diagnosis (T0) and on occasion of last follow-up (Tx). * significance *p* < 0.05, ** *p* < 0.005, *** *p* < 0.0005, **** *p* < 0.00005. Ct: centile, Hb: haemoglobin, HbA1C%: glycated haemoglobin, ALP: alkaline phosphatase, PTH: parathormone, HDL: high density lipoprotein.

**Table 1 nutrients-16-01708-t001:** Epidemiologic features of the study cohort.

	N	Sex	Age at Diagnosis (years)	Follow-Up Duration (years)	Familiarity for CD	Marsh
Whole cohort	171	F 60%	6 (1.6–17)	3 (0.35–15.3)	Yes 35%	0–50% 1–50%
Only anti-Tg2 persistently + patients	134	F 57%	5.9 (1.6–17)	3.19 (0.35–15.3)	Yes 36%	0–46% 1–54%
Only patients with a follow-up > 3 years	75	F 65%	5.7 (1.5–13.3)	5.7 (3.01–15.3)	Yes 36%	0–43% 1–57%

**Table 2 nutrients-16-01708-t002:** Summarised results of subgroup analysis. ALP: alkaline phosphatase, PTH: parathormone, Ns. not significant, ULN: upper limit of normality.

		Only Anti-Tg2 Persistently Seropositive Patients (N = 134, 78%)			Only Patients with a Follow-Up > 3 Years (N = 75, 44%)		
		Mean Value T0	Mean Value Tx	*p*-Value	Mean Value T0	Mean Value Tx	*p*-Value
Anti-Tg2 and gluten intake	Anti-Tg (x ULN)	3.9	2.2	<0.005	4.2	1.9	<0.005
	Gluten (g/day)	23	21	0.2	22.8	21.5	0.47
Growth	Height (ct)	50.5	48.6	0.25	54.08	49.2	0.05
	Weight (ct)	47.2	50.6	0.09	50.01	55.7	0.11
	BMI (ct)	49.7	51.3	0.49	49.7	59.2	0.01
Iron metabolism	Hb (mg/dL)	12.8	13.3	<0.005	12.8	13.4	<0.005
	Blood iron (µg/dL)	81.6	90.8	0.03	81.7	95.6	0.02
	Ferritin (ng/mL)	34.4	35	0.8	36.8	37.8	0.76
Glycemic metabolism	HbA1C (%)	5.27	5.3	0.08	5.2	5.3	<0.005
	Glycaemia (mg/dL)	67.3	70	0.008	68.3	70.9	0.07
Nutritional status	Albumin (g/dL)	4.57	4.59	0.43	4.6	4.6	0.14
	Cholesterol (mg/dL)	154.92	154.99	0.97	162.6	157.7	0.15
	HDL (mg/dL)	54.3	57.7	0.01	53.3	55.1	0.46
Bone metabolism	Calcium (mg/dL)	9.8	9.7	0.06	9.9	9.65	<0.005
	Phosphorus (mg/dL)	4.8	4.5	<0.005	4.9	4.5	<0.005
	ALP (U/L)	229	218.6	0.16	226.3	215.3	0.44
	PTH (pg/mL)	41.9	45.7	0.23	42.4	46.8	0.17
	Vitamin D (ng/mL)	24.6	25.4	0.57	19.3	21.6	0.07
Thyroid autoantibodies	Anti-TG and Anti-TPO	9 positive (7%)	11 positive (8.2%)	Ns Ns	7 positive (9%)	11 positive (14%)	Ns Ns
Type 1 diabetes autoantibodies	Anti-IAA Anti-IA2 Anti-Zn-T8 Anti-GAD	5 positive (4%)	4 positive (3%)	Ns Ns Ns Ns	8 positive (10%)	4 positive (5%)	Ns Ns Ns Ns

## Data Availability

The original contributions presented in the study are included in the article, further inquiries can be directed to the corresponding author.

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
