# Peer review of "Potential Celiac Disease in Children: Health Status on A Long-Term Gluten-Containing Diet"

_nutrients, 2024, doi:10.3390/nu16111708_

Round 1

Reviewer 1 Report

Comments and Suggestions for Authors

Manuscript ID nutrients-3012056

Article Title: Potential celiac disease in children: health status on a long-term gluten-containing diet

Congratulations to the Authors for writing a paper on a topic that is very important from a clinician's perspective. However, I would like to add some comments and suggestions:

MATERIALS AND METHODS

1.      In verses 80-81 authors wrote that  … 119/443 (27%) abandoned the cohort because they permanently stopped producing antibodies despite being on a GCD.”  And in verses 97-98 they wrote:Since the cohort included also patients that had  permanently stopped producing antibodies over time (N=37) we run a second analysis excluding them”.

Please make this more clear in methodology section. May be a study flow chart would be helpful.

2.      What that means that some of studied children stopped producing antibodies PERMANENTLY, i.e.  how many times the tests for the presence of anti-TG or others celiac autoantibodies were negative ? For how long?

               RESULTS:

1.      Authors presented in this section changes in gluten intake during follow-up. But there is nothing about assessment of gluten intake in methodology section.

How was gluten intake assessed?

Please add this to the methodology.

2.      How Authors explain the fact that percentiles of BMI and weight increased during follow-up and percentile of height did not change.  In fact, Figure 1 and Table 2 suggests as if percentile of height was near being reduced (?) in follow-up (maybe I am wrong, please correct me).

In discussion Authors stated that they “could not find a worsening in the growth”.   Please explain this (extend this section in discussion). 

3.      In Figure 1 gluten intake should be presented in g/day (consequently as in text of Results). Please explain “ULN” in table 1 description.

4.       140-142There was not a statistically significant change in the number of patients that presented at least one positive autoantibody for thyroiditis (anti-TG and anti-TPO, p 0.06) or for type 1 diabetes (anti-IAA, anti-IA2, antiZn-T8, anti-GAD, p 0.15). ANA were detected 142 in 23 patients at the end of the period of follow-up, in no case at diagnosis.” 

I am eager to know how many patients (%) in study group presented other autoimmunities.  Maybe Table 2  or a new one would present these data ( N with % of group) in all analyzed groups?

5.      In table 2 instead of term NUTRITION I suggest NUTRITIONAL STATUS.

 Please add the number of studied in subgroups and percentages (not only N).

DISCUSSION:

1.      In my opinion it has no clinical sense to state that HbA1c level increased (being in the referenced value) in this group, (but I am not a pediatrician).

2.      The Authors compare their results with only one study [ref. 17] in adults with celiac disease. If there are no other observations of children with asymptomatic potential celiac disease it should be more emphasized in discussion.

3.      The discussion is a bit short. I suggest adding a few sentences about potential negative nutritional consequences of gluten free diet (which are important when this diet is introduced without proper indications).

Author Response

Attached hereafter

Reviewer 2 Report

Comments and Suggestions for Authors

Some minor comments on the manuscript “Potential celiac disease in children: health status on a long-term gluten-containing diet.

My main issues and comments relate to the section “study population and methods”.

From the original PCD cohort of 319 subjects – 171 (54%) were retrospectively selected based on the available data (clinical and autoimmune data). Is it reasonable to state that these subjects are a representative sample of the whole PCD cohort (n=319)?

The main cohort (n=171) subjects were investigated at two time points (at time of 1st biopsy and last visit of follow-up). The final/main analyses were performed on 134 subjects. The authors mention that subjects with iron and/or vitamin D supplementation were excluded from the analysis. How many subjects were there? This information is missing. Depending on the number of subjects, wouldn’t it be interesting to look at differences between subjects that were supplemented with vitamins/minerals compared to not? 37 subjects were excluded because they stopped producing antibodies. Were any of the subjects that were supplementing included in this group? This needs to be clarified.

Table 2 – for clarity – add the number of included subjects in the two columns “Only antiTg2 persistently seropositive patients” and “Only patients with a follow-up > 3 yrs”.

 Long-term gluten-containing diet. Among the 134 subjects that – how many were followed in less than 12 months? The range follow-up range is described to be 0.35 – 15.3 year. I wouldn’t call 4 months as long-term diet.  

I couldn’t access the paper describing original cohort (ref #12) – so I’m curious how gluten consumption was assessed? Food Frequency Questionnaires? Food records? The dietary assessment method should be mentioned somewhere. Also, if the subjects had additional clinic visits between the two measurement points – it would be interesting to see if the diet (gluten intake) was consistent during the whole period.

Still, even if the study has some issues regarding describing the study population, the results are interesting, especially the group followed for more than 3 years. The results are comforting regarding health parameters in this group.

Comments on the Quality of English Language

No further comments.

Author Response

attached hereafter
